# Fleet Management and Control System for Medium-Sized Cities Based in Intelligent Transportation Systems: From Review to Proposal in a City

**Beimar Rojas** [1], **Cristhian Bolaños** [1], **Ricardo Salazar-Cabrera** [1,*],
**Gustavo Ramírez-González** [1], **Álvaro Pachón de la Cruz** [2] **and**
**Juan Manuel Madrid Molina** [2]

1   Telematics Engineering Research Group (GIT), Telematics Department, Universidad del Cauca,
    Popayán 190001, Colombia; beimaro@unicauca.edu.co (B.R.); crissebas@unicauca.edu.co (C.B.);
    gramirez@unicauca.edu.co (G.R.-G.)
2   Information Technology and Telecommunications Research Group (I2T), ICT Department, Universidad Icesi,
    Cali 760001, Colombia; alvaro@icesi.edu.co (Á.P.d.l.C.); jmadrid@icesi.edu.co (J.M.M.M.)
*   Correspondence: ricardosalazarc@unicauca.edu.co; Tel.: +57-313-586-0304

**Abstract:** In medium-sized cities in developing countries, transit services without dedicated lanes have issues related to route compliance, schedules, speed control, and safety. An efficient way for dealing with this issue is the use of Information and Communication Technologies (ICT), to implement a Fleet Management and Control Systems (FMCS). Such implementation can be performed using Intelligent Transportation Systems (ITSs), which allow integration of services and adequate standardization. This article features: (a) a literature review, related to FMCS based on ITS and enabling technologies, (b) design of the ITS architecture of an FMCS, and (c) some advances in the development of the proposed FMCS in a Colombian city (Popayán). The results of the literature review allowed identifying the most important requirements of FMCS in order to design the ITS architecture and build a prototype featuring the suggested technologies. Finally, some experiments were performed to evaluate the operation of the developed prototype. The results showed evidence of adequate operation in sending and receiving messages from and to four prototypes developed for the vehicles, also complying with the established requirements of location, tracking, exchanged data, and security. This allows continuing the development of the proposed FMCS, with some adjustments.

**Keywords:** Intelligent Transportation Systems; ITS architecture; fleet management; transit vehicles; ITS enabling technologies

---

## 1. Introduction

Cities in developing countries such as Colombia face mobility problems, such as high traffic congestion on roads, high traffic accident rates, and high levels of air pollution and noise. Many of these problems are caused by public transportation [1,2].

Therefore, a system seeking to improve control and monitoring of transit vehicles for cities of Colombia and other Latin American countries could indirectly contribute to improving the mobility problems of such cities. Medium-sized cities (with populations between 100,000 and 1,000,000 inhabitants) use "collective" service as the principal transit service. "Collective" service is provided by small buses or vans with a capacity between 12 and 20 passengers, through public roads with other types of vehicles (e.g., private vehicles). The service has specific stops for passenger pick up/drop off, but a considerable

number of users and drivers do not respect this, stopping anywhere along their route. The "collective" service also has problems related to speeding, noncompliance with regulations, vehicle routes, schedules, and frequencies [3,4].

Information and Communication Technologies (ICT) emerge as the best option for the implementation of an efficient transport system because they allow centralized traffic control, navigation, security, access to travel information, and real-time assistance, among other services [5]. Such systems belong in this work to the context of the Intelligent Transportation Systems (ITSs) and Fleet Management and Control Systems (FMCSs) [6,7].

The Fleet Management Control Systems (FMCS) are responsible for controlling vehicle operation and evaluating compliance with scheduled services. An FMCS monitors vehicles in real-time and generates alerts when events happen. In general, Fleet Management Systems (FMS) reduce risks, increase the quality of service, and improve operational efficiency at a minimal cost [8].

The increase of quality in transit service implies providing a better service to users. The main need for users of "collective" public transport is to know the current location of the vehicle they are waiting for, to be able to roughly calculate the approximate time of arrival at a stop [9]. To determine the current location of the transit vehicle, it is necessary to continuously track its location (with the use of ICT); then, by factoring variables as actual traffic level and historic route time history, an approximate time of arrival at a certain stop can be calculated. Uncertainty about the timely arrival of the vehicle is one of the main reasons why the level of use of this means of transport is relatively low in times of low demand [7]. Hence, continuous monitoring of the transit vehicle is a very important requirement to consider in an FMCS, to improve the quality and usability of the transit service.

FMCS can improve operational efficiency in aspects such as fuel consumption, which is a considerable percentage of operational cost. Fuel consumption generates $CO_2$ emissions, which considerably affect the air quality of a city [10,11]. Fuel consumption is related to the way the vehicle is driven [12], so monitoring the speed of a vehicle on the route can help to identify inappropriate driving behaviors, e.g., running a red light [13]. Speed control is also an important factor for reducing risks, considering that one of the main causes of road accidents in the world is speeding. Therefore, speed control of the vehicle along its route is an important requirement of an FMCS, to reduce risks and decrease fuel consumption.

FMCS developed recently commonly use cellular data transfer (GSM, GPRS, or LTE) for communication between transit vehicles and control centers, which causes issues related to high operating costs, and difficulty in maintaining the system [14]. Some emerging communication technologies such as low-power wide-area networks (LPWANs) can improve upon some factors of cellular communication technologies, such as range, availability, frequency band, and cost reduction both in acquisition and operation [15]. The use of adequate communication technology between transit vehicles and the control centers is an important requirement for an FMCS to improve operational efficiency at a minimal cost.

It is also important that an FMCS is not complex in its handling and operation because this could mean more delays in processes, which could make the system slow and inefficient. That is why it is convenient to properly distribute the workload between components and operating personnel [16]. The simplicity of handling and division of work does not imply a lack of security because it is also necessary to include security components to guarantee confidentiality, integrity, privacy, and availability of the managed information [17,18].

Finally, it is important that an FMCS must also consider the safety of passengers because it is necessary to protect passengers' well-being, thus building trust in the system [19]. Speed control is a factor that contributes to increasing safety in the vehicles that are part of the FMCS. Proper vehicle tracking can also identify some events that may pose a safety risk for passengers, such as unscheduled routes or stops at prohibited sites.

Considering ITS developments, the significant disadvantage of most of the ITS services developed at the international and Colombian level is that they are not based on adequate architectures, which brings integration, interoperation, and reusability issues [20]. Many models of FMCS have been

proposed, but most of them are implemented with services not based on an appropriate architecture. Although there are several approaches to Colombian ITS architectures, the proposed modeling is very general and not suitable for cities with specific characteristics [21].

To address this situation with transit service in the mentioned context, this research focuses on:

- A literature review (with scientific mapping and systematic review) of FMCS.
- Proposal of an ITS architecture for the FMCS in medium-sized cities in the Latin American context, by previously identifying suitable requirements for an FMCS in such context. In addition to the ITS architecture, suggested technologies and a prototype were defined.
- Experiments for validating usage of the suggested technology and prototype in a case study (Popayán medium-sized Colombian city).

The article structure is as follows: Section 2 presents the methods and materials used in this work and the criteria for the selection of a communication technology; Section 3 presents the literature review with SciMAT tool [22], and Preferred Reporting Items for Systematic Reviews and Meta-Analyses (PRISMA) methodology [23]; Section 4 presents the identification of suitable requirements of an FMCS, a study of ITS reference architectures, a proposal for an ITS architecture for the FMCS, and a proposal for an FMCS prototype; Section 5 contains experiments performed to validate the proposal presented in the above section; Section 6 presents a discussion of the obtained results; finally, Section 7 presents the conclusions.

## 2. Materials and Methods

This section presents the methodologies, methods, processes, and materials used in this work.

### 2.1. Literature Review

Initially, a scientific mapping analysis was performed with the help of the tool called SciMAT [22], in order to analyze the social, conceptual, and intellectual evolution of the interest area. In this way, relevant aspects and topics were identified and included in the literature review. Then, a systematic review of the literature was carried out in some databases related to the research topic. The review sought to identify different ITS services and architectures used in FMCS, as well as the methods used for their development, the communication technologies, and the obtained results.

Several tools for bibliometric and scientometric studies are available. SciMAT was selected because it allows the analysis of scientific mapping within a longitudinal framework and provides different modules that help the analyst perform all the steps of the scientific mapping workflow.

The procedure for scientific mapping included the following steps:

- General Flow Settings. In this step, the configuration for mapping was determined. A query string was determined and used with the SCOPUS database (due to its compatibility with SciMAT and high worldwide recognition). In the definition of time segments to allow a longitudinal analysis and discover the conceptual evolution of the field of study, three consecutive periods were considered: 2014–2015, 2016–2017, 2018–2020.
- Analysis of Results. Strategic diagrams were initially generated to measure the performance, quality of the subject, and subject areas detected. Subsequently, a review of the evolution was performed, considering the number of keywords and the number of shared keywords in the different subperiods, and the thematic evolution of the research.
- Mapping conclusions. Relevant topics and other important aspects were identified to perform the systematic review.

To address some limitations found in the scientific mapping (related to the type of product and license used) with respect to the number of available databases and the impossibility of detailed identification of documents to find some comparison parameters, a systematic review was performed

using the Preferred Reporting Items for Systematic Reviews and Meta-Analyses (PRISMA) methodology. The phases of such review are detailed below:

- Identification phase. A bibliographic search was performed on the following topics: Intelligent Transportation System, communications, security, and Fleet Management and Control System (FMCS). The three selected databases were: IEEE Xplore Library, eBook Academic Collection (EBSCO), and ScienceDirect. The search was limited to documents published over the last five years. This search was concluded in March 2020. These databases were chosen in addition to SCOPUS (which was used in the scientific mapping) to achieve two objectives: (1) expand the scope of the review, thus enriching the model through different sources of information, and (2) to check the relationship between scientific mapping and systematic review.
- Screening phase. All the documents from the different sources were joined and duplicates were eliminated. Then, the abstract of each document was reviewed; those documents not showing a direct relationship to the purpose of the review were discarded.
- Eligibility phase. The remaining documents were fully reviewed and filtered according to in-depth criteria defined in the previous phases of the systematic review and the results obtained from the scientific mapping.
- Included phase. The documents were divided into four different groups in this phase. Subsequently, a qualitative synthesis was performed for all groups, and according to the parameters defined for each one, quantitative synthesis was made for the same four groups.

Finally, the scientific mapping and systematic review allowed identifying features essential to define the requirements of an FMCS in the context of a "collective" service for a medium-sized city.

## 2.2. Proposal of ITS Architecture for FMCS and Prototype

Initially, the requirements of an FMCS for the specific context were compiled from the literature review and analysis of the environment. Next, in order to find an adequate reference ITS architecture, the most widely used architectures worldwide were analyzed, identifying the essential aspects of each one. This allowed the creation of a proposal of ITS architecture for the FMCS in the specified context.

From the ITS architecture, a scope was determined for the development of the FMCS prototype. Once the scope of the prototype was defined, the technologies to be used in this prototype were selected. According to the results of the literature review, such technologies are related to the following topics:

- Transit vehicle location technology (considering speed measurement).
- Communication technology between the transit vehicles and Transit Management Center.
- Technology to guarantee the security of the information transmitted between vehicles and the Transit Management Center.
- Technology to guarantee the security between other modules of the system.

The systematic review showed that almost all works featuring vehicle tracking used Global Navigation Satellite System (GNSS) services for positioning, particularly the Global Positioning System (GPS) technology. Some other works recommend the use of Short-Range Wireless technologies (SRWT), but this option needs a large infrastructure and does not allow continuous tracking, leaving GNSS as the best option. GNSS technology also allows speed measurement, a key aspect for an FMCS.

The communication technology to be used between the transit vehicles and the Transit Management Center (TMC) is one of the decisions that should be discussed. Reviewing [24], an analysis of different communication technologies for FMCS was performed, including cellular network technology, Ubiquiti, radio trunking, low-power wide-area network (LPWAN), and SRWT such as Bluetooth, ZigBee, and WiFi. The most important parameters in a communication system such as coverage, costs (of implementation and operation), scalability (appropriate for the growth of the network infrastructure), low latency, and frequency were considered, yielding LPWAN as the best for an FMCS system. In addition, Reference [25,26] analyze LPWAN technologies such as SigFox, narrowband

Internet of Things (NB IoT), DA SH7 alliance protocol (D7A), Ingenu, Telensa, random phase multiple access (RPMA), and long range (LoRa). This review deemed LoRa [27] as the most appropriate for communication between vehicles and TMC, due to its high level of coverage; low implementation and operation costs; use of frequencies in the industrial, scientific, and medical (ISM) radio bands, and low latency.

LoRa is suitable for an FMCS between the mentioned modules because of the following features:

- Range (approximately 5 km in urban areas with line of sight).
- Transmission speed (37.5 kbps).
- Use in "free" frequencies (ISM), which do not require a spectrum use license.
- Its ability to send unlimited messages. Here it is important to highlight that the packet size of this technology is relatively small (255 bytes), however, its ability to send unlimited messages counters this weakness.
- Its status in the global market, achieving rapid growth for IoT communications, rivaling NB IoT as the best in the LPWAN field [28].
- The level of security. LoRa implements a handy protocol called "Long Range Wide-Area Network" (LoRaWAN), designed to wirelessly connect "things" to the internet with a bidirectional communication featuring end-to-end security [27].
- Data packets can be sent simultaneously, thus eliminating collision problems.

Regarding a technology to guarantee the security of the information transmitted between vehicles and the Transit Management Center, the LoRaWAN protocol is a good option. This protocol also allows point-to-point encryption of messages, ensuring that the information is not altered or eavesdropped by possible intruders. Finally, it guarantees that only information from registered devices is received, thus guaranteeing the integrity and confidentiality of the information [29].

The proposed architecture has different modules that must communicate through the network, so all communication must be done through the secure hypertext transfer protocol (HTTPS) which assures the information reaches only the intended recipient, and its headers send an authentication value with a key. In the same way, for the transmission of data, the secure file transfer protocol (FTPS) is used, and to guarantee the security of access to information, a hierarchy of users with limited permissions to obtain data is proposed.

Once the technologies were selected, the components of the prototype to be developed were defined. The module that was determined suitable for transit vehicles has the following principal components: (1) Heltec WiFi LoRa 32 (v2) device (which has a LoRa SX1276 chip, an OLED 0.96-inch display, and an ESP32 microcontroller unit or MCU); (2) Ublox 6M GPS module, to obtain vehicle location. These two components were integrated, achieving a low-level prototype. The selected LoRaWAN gateway was the Dragino LPS8 and the platform used was The Things Network (TTN).

### 2.3. Prototype Experiments for Validation

The proposed prototype was tested in a city used as a case study, the city of Popayán. Popayán is a medium-sized city in the southwest of Colombia (Latin America), it has an extension of 512 km$^2$ (including rural and urban areas) and a population of approximately 300,000 inhabitants. Considering the choice of LoRa as the communication technology for the prototype, a coverage analysis of this technology was performed for the city, using the QGIS (Quantum Geographic Information System) tool. QGIS was used because it has a plugin that allows the estimation of radio coverage taking into account demographic data and city buildings [30]. The study was performed in order to find an adequate location for the gateways and the total number of these devices that would be necessary to cover the city.

Once the coverage analysis was performed, a suitable location in the city was determined to develop the experiments. The objective of the experiments was to validate sending and receiving messages from and to the modules located in the transit vehicles (using four of these devices), determining

parameters such as signal level, distance reached, percentage of packets received, and operation of the data security layer.

Section 5 of this document explains in detail the design and results of experiments.

## 3. Literature Review Results

### 3.1. Scientific Mapping

The scientific mapping yielded the following results:

- General Flow Settings. The initial configuration for mapping yielded 1729 results, three of them were deleted due to format issues or replication. Once filtered by time period, 434 documents were included in the 2014–2015 category, 543 in the 2016–2017 category, and 749 in the 2018–2020 category.
- Analysis of Results. The strategic diagrams show that in all the studied subperiods, the emerging (wireless-telecommunication-systems, sustainable-development, Internet of Things) and driving (security, intelligent systems, ITS, ad hoc networks) topics achieved the highest citations scores and impacts. It was also possible to determine a series of trends in the characteristics and technologies used for the systems under investigation, including security and Intelligent Transport Systems (ITSs). Figure 1 presents the strategic diagrams of the three time periods considered, based on the number of published documents. Through the revision of the evolution, it was evident that three themes (security, ITS, wireless communication systems) found in the strategic diagrams are identified as relevant, confirming their importance for the systematic review.
- Mapping conclusions. The topics related to security, ITS, transit systems, wireless communication technologies, and IoT, should be deemed as relevant in the systematic review.

Only a few details about the process and the results of the scientific mapping are presented. Nevertheless, the authors can be consulted for any additional information or the generated diagrams.

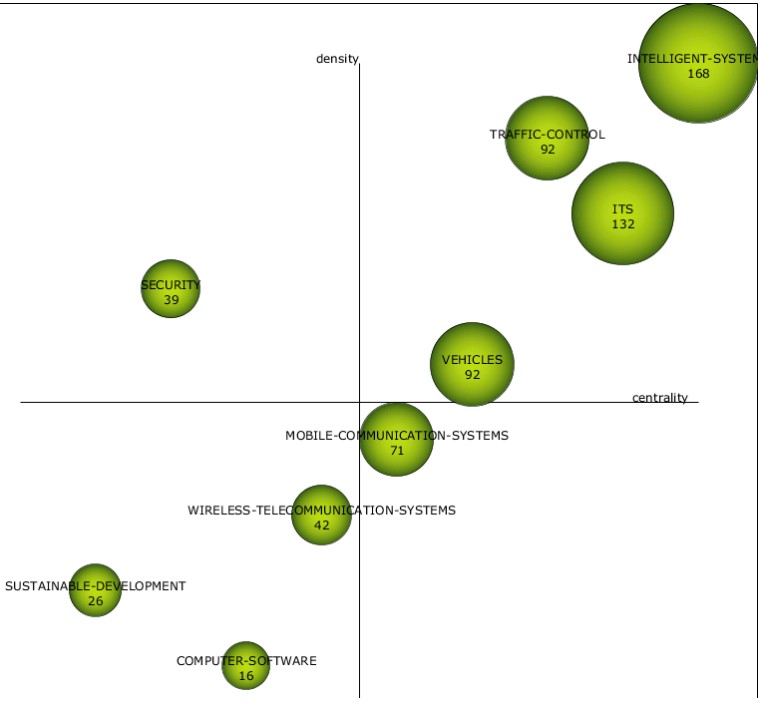

(**a**)

**Figure 1.** *Cont.*

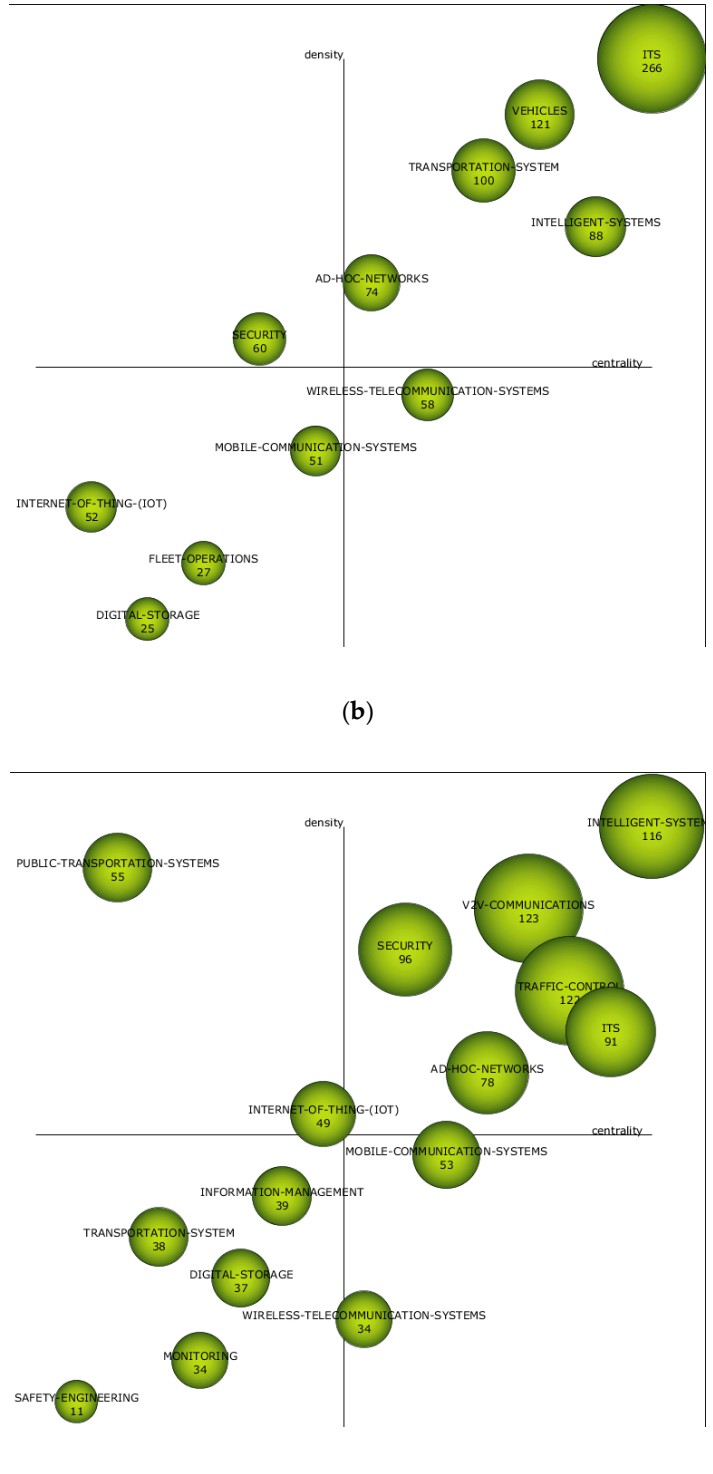

(**b**)

(**c**)

**Figure 1.** Strategic diagrams based on the number of documents published: (**a**) 2014–2015 time period; (**b**) 2016–2017 time period; (**c**) 2018–2020 time period.

## 3.2. Systematic Review

The systematic review was done using the Preferred Reporting Items for Systematic Reviews and Meta-Analyses (PRISMA) methodology. The searches were performed in the three databases mentioned in Section 2.1 (IEEE Xplore Library, EBSCO, and ScienceDirect). The documents found in such databases were identified using some search strings related to research terms, adding some recommended from

external sources. The initial search yielded a total of 2139 documents. These documents were evaluated and filtered according to the PRISMA methodology. In the final phase, 56 documents of interest were selected, and quantitative and qualitative analysis was performed for such documents.

Figure 2 shows the exclusion and inclusion results obtained in each of the phases, and the percentage of distribution of documents in each group.

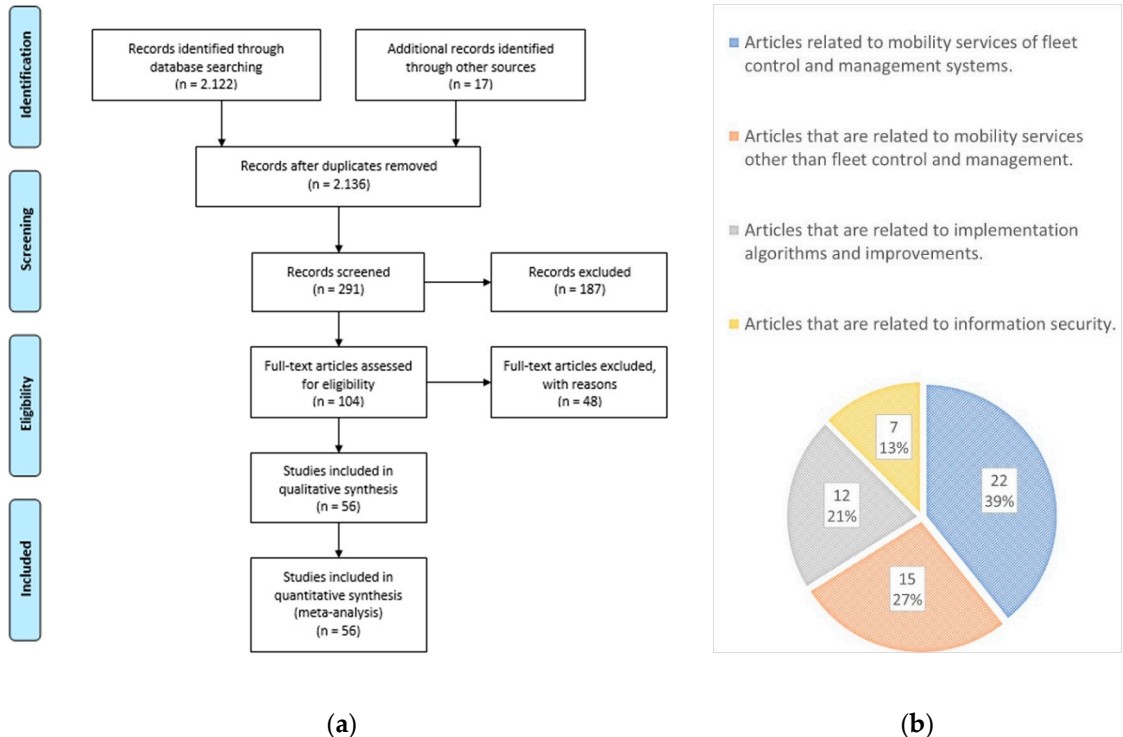

**Figure 2.** Results of the systematic review: (**a**) Preferred Reporting Items for Systematic Reviews and Meta-Analyses (PRISMA) Flow Diagram Results; (**b**) Classification by groups.

Groups identified in the included phase were: (1) articles related to mobility services of fleet control and management systems, (2) articles that are related to mobility services other than fleet control and management, (3) articles that are related to implementation algorithms and improvements, and (4) articles that are related to information security.

The following section presents only the results obtained in the last phase of the systematic review (quantitative and/or qualitative synthesis for each one of the four groups of documents).

### 3.2.1. Mobility Services of FMCS

Details of the documents reviewed in this group are shown in Table 1. From the total of articles in this group (56), 39% (22) are related to FMCSs. The quantitative study of this group was based on some pre-established parameters, such as the type of technology used for communications, the application environment, the use of security, the foundations of ITS, and the type of implementation. Accordingly, it was determined that none of the proposals made a full implementation of the defined parameters because many do not make use of ITS architectures or services (59%), little security implementation is seen (13%) and only a few are intended for implementation in Colombian or Latin American medium-sized cities (26%).

**Table 1.** Documents of mobility services of Fleet Management and Control Systems (FMCS).

| Article | Communication Technology | Based on ITS | Implementation Type | Applied Environment | Use of Security |
|---|---|---|---|---|---|
| Architecture based on open-source hardware and software for designing a real-time vehicle tracking device. (English) [31]. | Cellular network (GSM) | No | Real | Medium-sized city for Latin American context | No |
| Performance analysis of advanced bus information system using LTE antenna [32]. | Cellular network (LTE) | Yes | Not specified | Not specified | No |
| IoT-based predictive maintenance for fleet management [33]. | WiFi and cellular network | Yes | Real | Not specified | No |
| A Decision Support System based on smartphone probes as a tool to promote public transport [34]. | Cellular network (GSM, GPRS and UMTS) | Yes | Simulation | Small town for European context | No |
| TransMilenio: A High Capacity—Low-Cost Bus Rapid Transit System Developed for Bogotá, Colombia [35]. | Cellular network | No | Does not apply | Big city for Latin American context | No |
| Study on Real-time Bus Arrival Information System Based on Bluetooth [36]. | Bluetooth and cellular network | Yes | Not specified | Not specified | No |
| An Internet-of-Things-Enabled Connected Navigation System for Urban Bus Riders [37]. | WiFi and cellular network | Yes | Real | Big city for European context | No |
| Fleet Management and Control System from Intelligent Transportation Systems perspective [20]. | LoRa | Yes | Not specified | Medium-sized city for Latin American context | No |
| Data analysis and information security of an Internet of Things (IoT) intelligent transit system [38]. | WiFi | No | Real | Small town for American context | Yes |
| A low-cost M2M architecture for intelligent public transit [39]. | ZigBee and cellular network | No | Real | Not specified | No |
| Real-time vehicle fleet management and security system [40]. | Cellular network (GSM) | No | Real | Not specified | Yes |
| Monitoring System for Intelligent Transportation System Based in ZigBee [41]. | ZigBee | No | Simulation | Medium-sized city for Latin American context | No |
| IoT enabled intelligent bus transportation system [42]. | RFID and cellular network | No | Real (laboratory) | Not specified | No |
| Smart Bus Station-Passenger Information System [43]. | WiFi and cellular network | No | Real | Cities in a European context | No |
| GPS based Public Transport Arrival Time Prediction [44]. | Cellular network (GSM) | No | Real | Big city for Asian context | No |
| A New Framework of Intelligent Public Transportation System Based on the Internet of Things [45]. | ZigBee, Bluetooth, and cellular network (GSM, GPRS and 4G) | Yes | Real | Big city for Asian context | No |

**Table 1.** *Cont.*

| Article | Communication Technology | Based on ITS | Implementation Type | Applied Environment | Use of Security |
|---|---|---|---|---|---|
| Public Transport Vehicle Tracking Service for Intermediate Cities of Developing Countries, based on ITS Architecture using Internet of Things (IoT) [46]. | Cellular network and WiFi | Si | Real | Medium-sized city for Latin American context | Yes |
| A smart cost-effective public transportation system: An ingenious location tracking of public transit vehicles [47]. | RFID and Ethernet | No | Simulation | Not specified | No |
| Technological web platform for integrated public transport system (SITP) of the West Center Metropolitan Area in Colombia [48]. | Does not apply | No | Real | Medium-sized city for Latin American context | No |
| Integration of Smartphone and IoT for development of Smart Public Transportation System [49]. | Cellular network (3G) | No | Real | Not specified | No |
| Systematic Development of Intelligent Systems for Public Road Transport [50]. | WiFi | Yes | Real | Not specified | No |
| Analysis of Radio Technologies for the Transmission of Information to the User of a Fleet Control System [24]. | LoRa and Ethernet | No | Real | Medium-sized city for Latin American context | No |

Regarding the type of communication used to connect vehicle devices with the cloud server, it was found that the cellular network (GSM, GPRS, 3G, LTE) is the most used (71%), followed by WiFi (28%), ZigBee (14%), Bluetooth (9%), LoRa (9%), Ethernet (9%) and RFID (9%). (in some cases, more than one type of technology is used, so the percentages add up to more than 100%).

Works that implemented technologies such as WiFi, ZigBee, Bluetooth, and RFID also located the vehicle when it passed through a checkpoint or bus stop. Although this allows for a low-cost location service, it does not allow vehicle tracking in real-time.

Some of the works in this group recommended that the FMCS managed certain types of information between the Transit Management Center and transit vehicles. It is recommended that the Transit Management Center (module in charge of managing all transit in a city) sends to transit vehicle information regarding assigned routes, traffic alerts, partial or total route changes, and the scheduled route performance for each of the stops. In turn, it is recommended that each vehicle informs the Transit Management Center: its location, current number of passengers, and compliance with schedules.

### 3.2.2. Mobility Services other than FMCS

This group features 15 documents (27% of the total). See Table 2 for more details. In this case, a more diverse use of alternative communication technologies is seen, with cellular communication dropping to 46% of participation and technologies such as ZigBee, Bluetooth, and WiFi increasing to 20%. LoRa only is featured in two documents (13%). Regarding the mobility services on which the documents focus, 53% correspond to traffic service in general, 20% to cargo truck fleets, 13.4% to emergency vehicles, and 14.6% to bicycles and school transport. It is interesting to note that different mobility services are involving technologies for the management and control of their systems, thus confirming the need to develop a scalable and interoperable system.

**Table 2.** Documents of mobility services other than FMCS.

| Article | Communication Technology | Service | Based on ITS | Implementation Type | Applied Environment | Use of Security |
|---|---|---|---|---|---|---|
| Internet of Bikes: A DTN Protocol with Data Aggregation for Urban Data Collection [51]. | WiFi | Bicycle tracking | Yes | Simulation | Medium-sized city for European context | No |
| Heterogeneous Intelligent Transportation System in Quito city under the paradigm of the SWE-SOS standard and IoT notifications [52]. | Cellular network (GSM) | General traffic | Yes | Real | Medium-sized city for Latin American context | No |
| A vehicular network-based intelligent transport system for smart cities [53]. | WiFi | General traffic | Yes | Simulation | Cities for Asian context | Yes |
| Cloudthink: a scalable secure platform for mirroring transportation systems in the cloud [54]. | Cellular network (GPRS) | General traffic | Yes | Real | Not specified | Yes |
| A mobile platform system for driving assistance [55]. | Cellular network, Bluetooth and WiFi | General traffic | Yes | Not specified | Not specified | Yes |
| Fleet Management System for Truck Platoons—Generating an Optimum Route in Terms of Fuel Consumption [56]. | Cellular network | Cargo truck fleets | Yes | Not specified | Not specified | No |
| Fleet Management Cooperative Systems for Commercial Vehicles [57]. | Cellular network | Cargo truck fleets | Yes | Not specified | Not specified | No |
| Smart fleet monitoring system using Internet of Things (IoT) [58]. | Cellular network (GSM) | Truck fleets | No | Real | Not specified | No |
| Intelligent transportation system based on the principles of service-oriented architecture [59]. | Cellular network (3G & 4G) | Emergency vehicles | No | Real | Large-size city in the Asian context | Yes |
| Analysis of handshake time for Bluetooth communications to be implemented in vehicular environments [60]. | Bluetooth | General traffic | No | Real | Not specified | No |
| Field testing of Bluetooth and ZigBee technologies for vehicle-to-infrastructure applications [61]. | Bluetooth and ZigBee | General traffic | No | Not specified | Not specified | No |
| The Application of ZigBee Technology to the Intelligent Bus Query System [62]. | ZigBee | School transportation | No | Real | Not specified | No |
| A novel approach of using security-enabled ZigBee in vehicular communication [63]. | ZigBee Area Network | General traffic | Yes | Simulation | Not specified | Yes |

**Table 2.** *Cont.*

| Article | Communication Technology | Service | Based on ITS | Implementation Type | Applied Environment | Use of Security |
|---|---|---|---|---|---|---|
| i-Car System: A LoRa-based Low-Power Wide-Area Networks Vehicle Diagnostic System for Driving Safety [64]. | LoRa | General traffic | No | Simulation | Not specified | No |
| Traffic Light Mobility Management Prototype for Ambulances, Under the Smart City Concept [65]. | LoRa | Ambulance system | No | Real | Medium-sized city for Latin American context | No |

Regarding the use of security, there is also an increase in participation, reaching up to 33%, and greater use of ITS services or architectures (53%), although there is a reduction in the implementation of the proposals in real environments (63%).

### 3.2.3. Implementation of Algorithms and Improvements Related to FMCS

This group features 12 documents (21% of the total). There is a significant increase in the use of WiFi technology (50%) and a reduction to 25% of the use of cellular networks. Technologies like LoRa, ZigBee, TETRA, and Ultra-Wideband have a usage percentage of 12% for each, thus evidencing the growth of particular technologies for IoT.

Similarly, there is an increase in the use of ITS architectures or services (75%) and a considerable reduction in the level of application in real environments (37%). WiFi technology stands out in this group (because simulations are commonly performed with algorithms).

The types of implementation for which the algorithms are designed do not differ much in their participation because of 12 documents found in this group, those that focus on communications, ITS architectures, transmission capacity and generation of alerts have the same number of occurrences (corresponding to 17%); those related to routing and data management account for 8% of the total; and the one with the most participation is "data collection", which has 25%. Sending alerts is an aspect to consider regarding the application of the algorithms, because as identified in the first group of this systematic review, it is important to send alerts from the Transit Management Center to transit vehicles so that they adjust their routes and are able to comply with the established schedules.

The high percentage of documents related to "data collection", and the science mapping analysis (specifically the strategic diagrams for the period 2018–2020) show a close relationship to digital storage and monitoring, two emerging issues within the field of research. Such issues are highly relevant to FMCS, so they should be considered for the proposed system [21,66–70].

### 3.2.4. Security Applications Related to FMCS

This last group features seven documents. In this group, the context of security application stands out, which is mainly on communications (43%), and it also has a high use of architectures or services (ITS) and a low implementation for real environments (one document).

Based on these data and those of previous groups, security is an important issue within the context of the study, a conclusion that ratifies the results of the science mapping analysis and verifies the need to implement this topic on the development of the proposed system. Similarly, the relationship between ITS architectures and security is observed in 71% of the documents, which stresses the importance of this topic in correctly defined systems that achieve interoperability and scalability [71–73].

### 3.3. Literature Review Conclusions

Based on the presented information, the predominant technology to geographically locate transit vehicles continues to be GNSS because it allows continuous tracking. Some other technologies such

as ZigBee, WiFi, RFID, and Bluetooth can be used to locate vehicles, but only when it passes very close (a few meters) to a certain point where the reader device is located, for example at stops to pick up/drop off passengers.

Regarding the communication technology between the vehicles and the Transit Management Center, cellular communication technologies still prevail; however, some important emerging technologies were identified. LPWAN technologies have been evaluated in some documents, identifying LoRa as a promising technology (comparing it with technologies such as Sigfox or NBIoT), for its already mentioned features.

A gradual increase in the use of applications with security was also observed in the proposals for various fields of systems (communications, authentication, and information security), Although, in most cases, appropriate tests are not conducted to validate the proposals, this increase infers the need to implement security mechanisms in mobility services systems.

The security applications in the analyzed documents focus on various aspects. Authentication of vehicles is one of them; various strategies are proposed to reduce time and improve the efficiency of this function. Other proposals focus on communication security, proposing improvement algorithms on the security protocols such as LTE, emphasizing the need to improve security requirements such as confidentiality, integrity, privacy, availability, and nonrepudiation. Compliance with these requirements is important for the exchange of data between the vehicles since, otherwise, the information could be vulnerable to external attacks and inappropriate handling. It is also important to highlight security in access to the system, guaranteeing that unauthorized parties are not allowed to access the system. The issue of anonymity must also be considered in the security component. Travelers, for example, do not need to identify into the system to obtain certain information; while other actors, such as vehicles and operators, need to identify themselves in order to monitor their actions.

## 4. ITS Architecture Proposed for the FMCS

### 4.1. Identification of FMCS Requirements

According to the literature review, the following aspects were identified in a development validation of an FMCS in the aforementioned contexts:

- The location of the transit vehicle in an FMCS is mainly done through GNSS technology. Although there is evidence of other technologies used for location (Bluetooth, WiFi, RFID), they do not allow continuous tracking, only detecting the arrival of the vehicle at stops, due to the limited range they have.
- Controlling the speed of the transit vehicle is a relevant factor in identifying accident risks, complying with traffic regulations, and identifying inappropriate driving behaviors. The use of GNSS technology for location has the additional advantage of measuring the speed of the vehicle on the route.
- Communication technologies for mobility services such as fleet management, applied in the context of this work, must also involve low costs (due to budget constraints these types of cities have). Aspects such as: wide range operation, sufficient data rate, sufficient bandwidth, licensing of the frequency spectrum used, and suitable packet size must be taken into account.
- Most recent cases use the LPWAN LoRa technology as a means for communication between vehicles and Transit Management Center.
- Few cases consider information security. However, this feature should be taken into account in the implementation of these systems, for guaranteeing trust and integrity. The information security in the system must be evaluated from different fronts, one is integrity and authentication between the vehicles and the Transit Management Center; another is user authentication for system managers; finally, exchange of information between modules is a key issue because there is no manual authentication process, nevertheless, the use of REST APIs mitigates this problem by means of a transport layer security (TLS) encryption. The HTTPS protocol also allows the integration of

authentication headers. There are different authentication methods, the one recommended in this proposal is based on an API key, which also allows configuring the authorization to the information (providing limited access to resources), also offering the possibility of performing analysis and supervision of who uses the API, when and how.

- Most of the implemented security schemes are related to vehicle communications and authentication. Regarding vehicle communications, some proposals improve upon aspects such as reliability, operational range, transmission rates, and attack prevention. In authentication, the focus is towards the identification and access of system users, which in turn enable validity in the information they transmit and receive.

- The exchanged data between vehicles and the system control modules in an FMCS must allow the vehicle to send data such as its location, arrival at assigned stops, and in some cases, the number of passengers. In turn, the vehicle must receive the information about the assigned route, the changes in this route, alerts, and assigned schedules.

- In an FMCS the vehicle must receive information sent from the Traffic Management Center, so the communication technology must implement "downlink" messages to facilitate this process.

- The use of suitable ITS architectures and their related services facilitate scalability and interoperability with other services. In most of the studied cases, these principles are taken into account; however, a large percentage of such cases do not correspond to architectures or services that are appropriate to a particular context.

Summarizing, the FMCS should comply with the requirements presented in Table 3.

**Table 3.** Requirements determined for the FMCS.

| Requirement | Related to | Proposal | Observation |
|---|---|---|---|
| Location technology of transit vehicles must allow continuous and efficient monitoring. | Transit vehicle location | Use of GNSS for location. | Although there are some alternative location technologies, these would not allow continuous vehicle tracking. |
| FMCS must allow speed control of the transit vehicle | Transit vehicle speed | Use of GNSS to measure speed, use of speed alerts and detection of inappropriate driving behaviors | Through speed alerts, the risks of accidents can be reduced. By detecting inappropriate driving behaviors, steps can be taken to reduce fuel consumption. |
| FMCS must allow interoperability and integration (with other systems or services) in the context where it is implemented. | ITS framework | Use of adequate ITS architecture for FMCS in the specific context. | An appropriate ITS architecture should be proposed, reviewing relevant references, and considering the context. |
| FMCS must use an adequate communication technology for the application context | Communication technology | Use of LoRa technology (using the network protocol LoRaWAN) | LoRa offers some notable advantages over other technologies for the specific service. LoRaWAN protocol can be a viable option that improves communication security and efficiency. |
| The exchange of data between the FMCS modules must be secure. | Information security | Secure information exchange (through a network protocol such as LoRaWAN, HTTPS, TCP). Access to the data with SHA passwords and user hierarchy. | LoRaWAN protocol will be of great help by providing the possibility of encoding messages and validation of devices in vehicles. Regarding the information that circulates through the software network, the TCP and HTTPS protocols will guarantee the secure exchange of information. Finally, each user will be given a password to access the system and a user hierarchy will be implemented. |

**Table 3.** *Cont.*

| Requirement | Related to | Proposal | Observation |
|---|---|---|---|
| Exchanged data between the vehicles and the control modules must allow permanent updating of the information in both directions, to provide a better service. | Exchanged data between modules | Transit vehicles send their location, arrival at assigned stops, and the number of passengers. The vehicles receive the assigned route, the changes in this route, alerts, and assigned schedule. | An FMCS is more than a tracking service. All required information regarding schedules, routes, changes, and alerts should be considered. |

*4.2. Reference ITS Architectures Review*

An ITS architecture offers a vision of what an Intelligent Transport System will look like from a system design perspective. There are three pioneering world reference architectures: ISO 14813 [74], ARC-IT [75], and Framework Architecture Made for Europe (FRAME) [76], which establish certain groups of transport services. An ITS architecture can be presented at several detail levels, depending on the scope of the system. A review of the FMCS-related services proposed each reference architecture follows.

4.2.1. ARC-IT (Architecture Reference for Cooperative and Intelligent Transportation)

This American architecture establishes 12 areas of transportation services [77]. The two ARC-IT services of interest for this work are Public Transportation Transit Vehicle Tracking, and Transit Fleet Management (abbreviated as PT01 and PT06). The first one focuses on monitoring the current location of the transit vehicle using an Automated Vehicle Location System, while PT06 allows scheduling and automatic traffic maintenance monitoring.

Both architectures keep certain similarities in their design because they share several of the physical objects that compose them, such as the Transit Management Center (TMC), in charge of managing the fleet of transit vehicles, and providing operations, maintenance, customer information, planning, and management functions for transit property in coordination with other modes and transport services.

4.2.2. Framework Architecture Made for Europe (FRAME)

FRAME establishes 10 areas of transport services. The Manage Public Transport Operations area is related to the FMCS service and includes seven high-level functions. Each function encompasses several specific fleet management services, including general programming of services and generation of information that can be made available to travelers [78]. Among these functions, four were identified as appropriate for the context of this study: Monitor PT Fleet, Plan PT Service, Control PT Fleet, and Provide On-Demand Services.

4.2.3. ITS Architectures Initiatives in Colombia

In Colombia, there is no established ITS architecture that serves as a model for the subsequent development and interoperability of ITS services, although some ITS services have been established in the past. Work is currently underway to adapt the services established solely by the ISO 14813 standard [79], for achieving consistency in the national ITS plan.

In 2010, the Colombian government developed a national ITS architecture based on the ARC-IT architecture, to be implemented in different cities in Colombia; however, it has never been implemented [80].

A proposal related to the design of an FMCS from the perspective of ITS was also revised [20]. Such a proposal is based on American architecture (ARC-IT). Due to its relationship with the current work, the architecture proposed in the following section took into account the architecture presented in that work.

### 4.3. Proposed ITS Architecture

According to the obtained and analyzed information, the ARC-IT American architecture was taken as a reference, due to the level of detail it presents for each service, which facilitates their development. Additionally, when the service areas of the different architectures are compared, ARC-IT is the one with the largest covered areas, which gives it a higher scope factor. However, this architecture must include the context of medium-sized cities in developing countries. The ITS architecture must also take into account the aspects identified in the systematic review as important requirements in this type of system (FMCS). One of the most critical requirements is security for communications, reliability, authenticity, among others. Another relevant requirement is the measurement of vehicle parameters and the choice of reliable, affordable communication technologies.

The proposed architecture design is presented in Figure 3. The architecture is presented using ARC-IT notation. In addition, Table 4 briefly explains the flow of information in the proposed architecture.

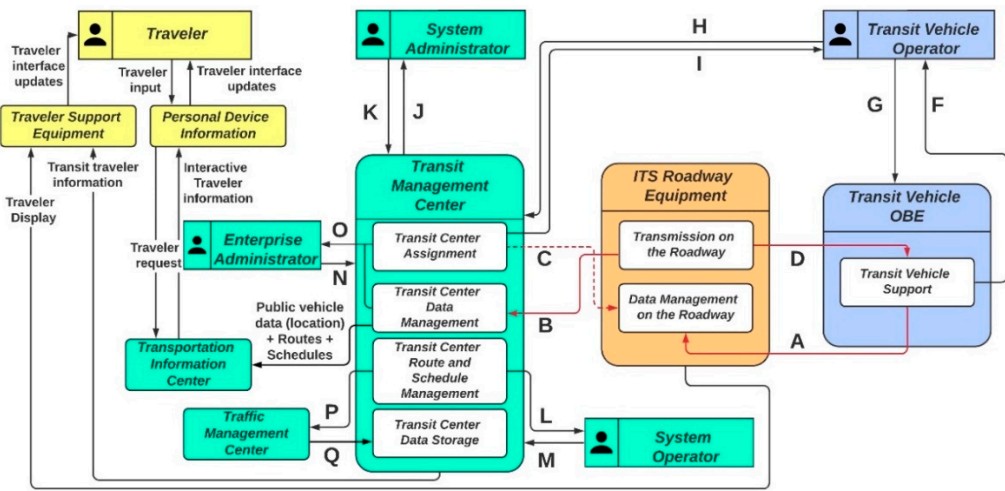

**Figure 3.** Intelligent Transportation System (ITS) architecture proposal for FMCS.

This architecture features a series of submodules (functional objects) for the Transit Management Center TMC and Transit Vehicle OBE physical objects. The ITS Roadway Equipment, with functionalities for the equipment on the road, is also included. Finally, two new actors (System Administrator and Enterprise Administrator) and new communication information were added. Thus, the ARC-IT base is preserved, keeping the Transit Vehicle OBE and Transit Management Center objects, and the Public Vehicle Operator actor together with most of the initial functions. It also includes several of the high-level functions of FRAME's Manage Public Transport Operations area, such as operational functions, continuous monitoring, continuous data collection, routes, schedule management, and provision of information for travelers, among others. Additionally, the on-demand service provisioning function of FRAME's Provide On-demand Services area was added.

To keep the architecture fully functional and scalable, communication messages with the other modules are defined so that information can be shared between them. To comply with already mentioned security requirements, the messages that go through the "Transit Vehicle OBE" and "TMC" modules will need layers of security to guarantee confidentiality, integrity, and availability of the information. Messages requiring security are marked with red lines in Figure 3. Likewise, authentication methods are applied for all information communication services between actors and modules.

It is important to emphasize that this architecture is generalizable to other medium-sized cities that handle the same context of the public transportation system ("collective" service). The use of an adequate architecture will allow the city to allow interoperability with other systems, in the same way, it facilitates the integration of new services that can be implemented to the system through modules and messages.

**Table 4.** Information flow of the ITS architecture proposed.

| Flow | Name | Description |
| --- | --- | --- |
| A | Location data + Vehicle information + Public vehicle schedule performance | Measured parameters in vehicles. |
| B | Location data + Vehicle information + Public vehicle schedule performance | The same information collected by the OBE is verified and passed to the Transit Center Data Management. |
| C | Public schedule info + Public operator info + Route + Alerts + Addressee | Assignment information to Transit Vehicle Support, this includes to identify the recipient. |
| D | Public schedule info + Public operator info + Route + Alerts | Information on schedules and routes assigned to the corresponding vehicle that requests it. |
| F | Public vehicle operator display | Information on the vehicle operator's work parameters, such as the assigned route number, schedules for that route, and alerts. |
| G | Public vehicle operator control | Manual interaction to operate the OBE device. |
| H | Public vehicle operator input | Data the vehicle operator sends to the system. |
| I | Route assignment + Vehicle assignment + Change alerts | Detailed information on the vehicle operator's work parameters, also includes detailed alerts on work parameter changes. |
| J | System information | Information generated each time the administrator has made a request to the system. |
| K | System administrator input | Access parameters of the system administrator and information request. |
| L | Public operation status + Traffic alerts + Suggestions | Information related to traffic alerts generated and suggestions for changes in routes and schedules. |
| M | System Operator input | Input or access data for the system operator. |
| N | Enterprise administrator input | Access parameters and requests information for the company administrator. |
| O | Enterprise information + change request | Responses to requests made by the company administrator. |
| P | Public vehicle data (speed and location) + Public vehicle conditions + Service demand quantity | Information related to different parameters measured and processed of the vehicles. |
| Q | Traffic images + Road conditions + Incidents | Predefined messages sent by "Traffic Management Center" to detect different traffic conditions. |

*4.4. FMCS Prototype Network Diagram*

After establishing the ITS architecture proposal for the FMCS, the scope of the prototype to be developed must be defined. A network diagram, with the modules selected for the prototype and the proposed communication technologies for the architecture, is presented in Figure 4. The prototype focuses on the three main modules of the architecture presented in Figure 3: TMC, ITS Roadway Equipment, and Transit Vehicle OBE.

Figure 4 shows the path followed by the information between the different modules, and also shows module location. The OBE is located in each vehicle, which communicates with the ITS Roadway Equipment located in the city buildings, in which a gateway is located.

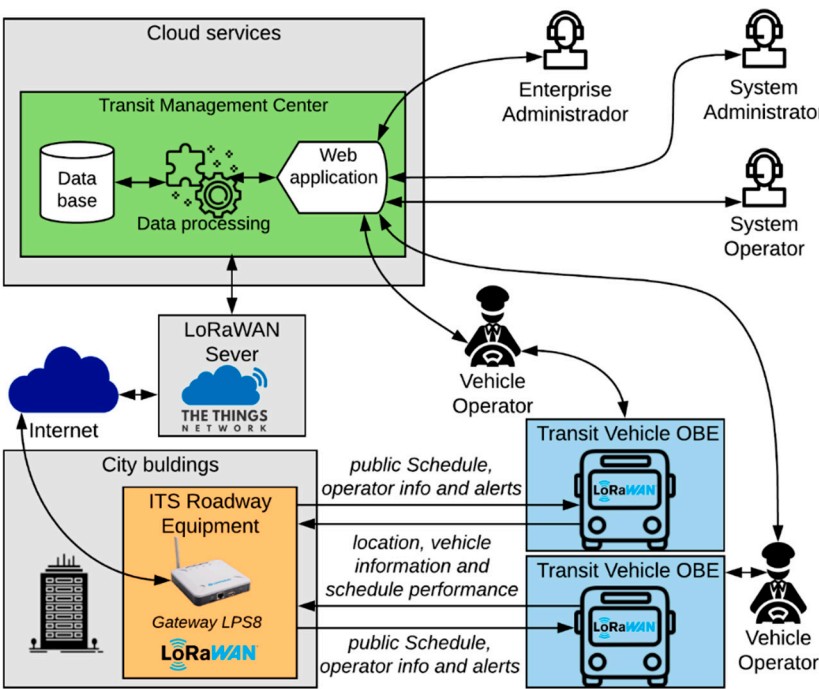

**Figure 4.** Network diagram for the proposed FMCS prototype.

The selected communication technology between Transit Vehicle OBE and ITS Roadway Equipment was LoRa, using the network protocol LoRaWAN. This was preferred over LoRa messages for two reasons. The first is that LoRaWAN provides an adequate security layer, the information is encrypted and the devices must join the network using the Over-the-Air Activation (OTAA) method in order to start sending messages. The join operation has several authentication parameters. The second reason is to avoid packet collision at the gateway (problem identified in the systematic review [14,81–83]) as LoRaWAN network protocol facilitates the dynamic change of LoRa communication parameters to minimize the probability of packet collision.

The use of the LoRaWAN network protocol requires the use of a LoRaWAN server, which is in charge of all the upper layers of communication; the proposed prototype uses The Things Network (TTN) platform [84]. TTN provides the possibility of sending the information coming from the LoRaWAN devices to servers in the cloud via the HTTPS protocol; this functionality was used to store the data received from the OBEs in a database. The code used for the Heltec cards (which includes integration with the GNSS module, the Ublox 6M GPS), the configuration of the Gateway Dragino LPS8 (mainly the setting of the adequate channels) and the configuration of the TTN platform (including the codification and decodification of data, the integration with a cloud server for sending data and downlink messages) is available upon request.

The next section describes analyses and experiments performed with the communication technology used between the OBE and the ITS Roadway Equipment, in order to determine the feasibility of the FMCS prototype.

## 5. Design and Results of Prototype Experiments

The coverage area analysis determined that a total of 13 gateways was necessary to cover the whole city of Popayán.

The experiments were developed to evaluate the proper functioning of the developed OBE devices, the selected gateways, the LoRa wireless communication technology, and the security of LoRaWAN protocol. The experiment used four OBE modules and one gateway. The location of the gateway was fixed, and the OBEs were able to move between 10 and 100 m of the gateway, all of them transmitting

at the same time. To verify operation at greater distances, the OBEs were subsequently located at distances between 100 and 400 m.

The obtained results were satisfactory, managing to receive 100% of the packets transmitted simultaneously in all the tests performed with Line of Sight (LoS) or minor obstacles. It was possible to visualize the online data sent from the OBEs to the gateway thanks to the TTN platform.

Problems arose with the presence of large obstacles, where the RSSI decreased significantly, thus receiving only 10% of the packets at a distance of 80 m. In this type of test with large obstacles, the maximum distance between devices was approximately 50 m to receive 90% of the packets. This is the reason why the gateway devices should be located in an area with few obstacles and at an elevated location. It is estimated that if 80% of packets are received correctly, the system would operate correctly.

The security of the LoRaWAN protocol was also verified. The joining process that the devices must execute to start sending messages with payload has the adequate security parameters (which are configured both in the code of the microcontroller card, and in the LoRaWAN server platform, TTN). Security tests were performed, trying to communicate a correctly configured gateway, with devices that were not correctly configured (with the LoRaWAN security parameters in OTAA mode), in these cases, the gateway detects the request, but does not allow the joining of the device, therefore it does not continue to receive packets with data. A security test was also executed, trying to connect a gateway that was not properly configured, in that case, it is not possible to join the devices (they had the correct security settings) and packets with data are not received from the devices.

The proposed FMCS requires duplex communication to send messages to and from the modules located in the transit vehicles, so tests of sending downlink messages were performed. In these cases, the packets are received at certain specific times by the device in the vehicle because the program that runs sequentially sends the sensed data at a certain point and receives the receipt confirmation (ACK). Once the data has been sent, it verifies the reception of "downlink" type messages. These delays detected in the reception of messages in the vehicle module could be considered thanks to the performed tests.

Finally, the tests related to the consumption of services through the TTN platform were successful. The server correctly stored the information received by the gateway. It is important to mention that this platform limits access only to authorized users.

With the results obtained in the experiments, it was possible to show that the requirements of the FMCS related to location, tracking, exchanged data, and security are met for the proposed design. However, it is important to clarify that the experiments were performed with a prototype of the system, which has a limited scope of FMCS.

## 6. Discussion

The literature review allowed identifying an overview of the approach for research on transport services, where ITS architectures or services and issues related to security are increasingly taken into account. This review allowed the identification of specific parameters in the FMCS, according to a considerable number of references from different databases in the last five years. The need to implement communication technologies alternative to cellular service was established (reviewing the identified parameters), which implies lower costs and probably better operation. The use of an ITS architecture was also identified as relevant because although many documents related to FMCS and other mobility services and/or systems mention the use of ITS or its services, almost none use an ITS architecture as the basis for their developments, which considerably makes difficult to standardize and integrate services. Another critical aspect identified in the literature review was the low number of proposed mobility services (especially FMCS), which had validation tests in real settings. This is very important because key aspects of operation can be identified and improved in application environments.

In addition, it was possible to identify the suitable requirements of an FMCS in the context of a "collective" transit service in a medium-sized city. These requirements can be considered relevant in the development of this type of system for Latin American cities that have similar characteristics.

In the case of cities with other characteristics, the process performed in this work to identify relevant aspects in FMCS can be applied in order to obtain results for the new context under study.

To propose the architecture of the FMCS, the principal reference architectures that exist worldwide were taken into account, specifically in proposals on public transportation. In addition, the requirements identified in the review for the context of the study and other related previous initiatives were considered.

The inclusion of the LoRa technology and its LoRaWAN network protocol was considered suitable for communications in the designed architecture, due to its particular characteristics that reduce operation and implementation costs. It also features a wide range of coverage with low power consumption and can be deployed easily.

There are some limitations regarding the proposed idea, specifically in the type of communication technology (LoRa) between OBEs and gateways. One of the most significant issues is the range limitation in places where there is no Line of Sight (LoS) and large obstacles between the different devices, reducing the number of packets received. Therefore, the location of the gateway in an elevated place is a relevant aspect. Another issue is the LoRa message size, which is small (255 bytes), however, this is not critical because the service works well with a low data rate, and additionally, the information sent can be encrypted, minimizing security risks. Regarding the maximum number of devices, although theoretically, a LoRa gateway can support up to one million nodes, it is important to perform scalability tests to validate this because the frequency of sending packets in an FMCS can affect this aspect.

Sending alerts to the vehicles is another important aspect in terms of limitations. The alerts must be handled in a relatively short time, so if the number of devices increases greatly, it will be difficult to handle this type of request due to the large amount of broadcasted information.

Rain can also significantly increase the level of losses in transmission levels, therefore it is recommended to factor this type of loss when defining the coverage area. Finally, the operating temperatures of the electronic elements used in OBEs are important aspects to consider as a limitation.

The proposed architecture has been designed to be implemented in medium-sized cities (100,000 to 1,000,000 inhabitants) that have a public transport system similar to the "collective" (buses of 12 to 20 passengers) and road infrastructure shared with private vehicles. It is important to consider these design aspects and the limitations (or restrictions) of the system, if this proposed architecture is going to be implemented in a similar city. For example, the geographical characteristics of the city must allow the correct location of the gateway devices to provide an optimal power level for transmission and an adequate LoS.

Regarding GNSS, tests showed that this type of technology is adequate to continuously determine the location of the vehicle with an error between 1 and 5 m, however, due to high power consumption and a long time to connect to satellites, a study of the different GNSS devices should be conducted to find the most suitable one.

The results obtained with the performed experiments show that the developed prototype meets certain requirements set out for the FMCS, however, its scope in functionality must be increased and tests with a greater number of nodes in an urban environment as close as possible to the real environment must be conducted for a more complete validation.

Finally, this proposal could be taken into account by local and national authorities related to the Strategic Public Transport Systems (SPTS) that are currently being implemented in eight cities in Colombia. The majority of these projects have not yet developed their technological components (FMCS among them), therefore, new requirements could still be included in the analysis of this system, related to requirements identified for FMCS, the ITS architecture, and the developed prototype. Considering these aspects, important benefits could be obtained in terms of system efficiency, standardization, and savings in implementation and operation costs.

## 7. Conclusions

With the performed literature review, the principal requirements of an FMCS applied in the context of medium-sized cities (for the "collective" transit service, which is common in many countries

of Latin America) were identified, such as interoperability and integration with other transport services, the use of adequate location, tracking, and communication technologies, and the measurement of suitable parameters for analysis and implementation of security protocols that guarantee the correct flow of information.

The design of the proposed architecture was done through a decentralized work approach, by dividing the operations of the system between different modules and actors, yielding a better distribution of processes and a decrease in the workload.

LoRa technology, with its network protocol LoRaWAN, was a good option as a communication technology between modules for the designed ITS architecture because it meets the requirements of low cost, good performance, and an adequate level of security, identified as necessary for the implementation of such architecture. Tests performed with LoRa technology (only LoRa packets) showed problems when two or more devices transmitted simultaneously, so the implementation of the LoRaWAN protocol was necessary, thus achieving effective uplink and downlink communication.

With the results obtained, it is feasible to continue with the development of the FMCS, by making some adjustments to the designed, developed and tested prototype to improve its operation, increase its functionality and verify its operation by scaling some parameters. An important aspect to improve the operation is to evaluate the possibility of changing the employed GNSS module, to improve the connection time to the satellite network and the energy consumption. To increase functionality, it is necessary to advance in the development of operations performed by all types of actors in the system. Finally, to validate the operation by scaling some aspects, tests have to be performed at a larger scale (with more gateways, more nodes, and larger distances), in order to guarantee the correct operation of the proposed architecture in real work environments (using vehicles on the roadways).

**Funding:** This research received no external funding.

**Acknowledgments:** Authors wish to thank Universidad del Cauca (Telematics Department) and Universidad Icesi (ICT Department) for supporting this research.

**Conflicts of Interest:** The authors declare no conflict of interest.

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
