# Peer review of "Fleet Management and Control System for Medium-Sized Cities Based in Intelligent Transportation Systems: From Review to Proposal in a City"

_electronics, doi:10.3390/electronics9091383_

Round 1
Reviewer 1 Report
In this paper, the authors have proposed a design of the ITS architecture of an FMCS. The study is very interesting. However, several concerns in the current stage of manuscript need to be addressed before consideration of acceptance, as follows:
To make the manuscript self-contained and more readable, make sure all the context and variables have the same font.
Please add more result or verifications for the proposed method.
More discussions and literatures should be added in the introduction.
Is there any other limitation of the proposed idea? Such as the specific or limited ranges of working environment or conditions? Is the proposed architecture still valid under other environment?
Carefully recheck grammar and typo errors.
Author Response
"Please see the attachment."

Reviewer 2 Report
Review of Fleet Management and Control System for Medium-2 sized Cities Based in Intelligent Transportation 3 Systems: From Review to Proposal in a City, MDPI Electronics 2020
This paper is about the development of a fleet management and control system (FMCS). A systematic literature review is done and the architecture and features of the system are adapted accordingly. A focus is put on communication technology.
Overall
- The paper IMHO tries too much. It is a review paper; at the same time proposes a reference architecture and implementation (with a very focused section on LoRaWAN – see section 5 analysis).
- I would recommend focusing on the review part; extend it (with more sources), make the requirements more explicit and then re-submit the paper. In its current form it cannot be accepted and needs a major re-write. I would encourage the authors to do that! They demonstrate good knowledge of the field.
Comments:
Line 30ff: The introduction is rather broad with many themes such us mobility problems, costs, etc. I miss a bit a clear description of the requirements (only in line 258 I can read the structure as a result of the review: communication, ITS architecture, …). Why wouldn't you talk about things like usability? Uptake? Costs? etc. in some detail at this point of the paper?
Line 180ff: the strong focus on the net coverage is somehow irritating. Why is that so important? Which other technologies (other than Lora) could you have used? What are the criteria for selection?
Line 224: I would have thought that typically in a systematic review you find many more documents (10.000 +) and you then iteratively narrow them down to a 100 or so (104 in your case) to actually read those. Why did you overall only find 291? Were the search strings defined too narrow?
Line 304, review conclusions: Mmmh – I am not sure whether I can follow you here. For instance, you argue with “Scaling” (line 318) which has not been part of the criteria you proposed?!
Line 342: you talk about security, which is important. True! Are you also addressing anonymity? Would that be another aspect? Or part of the (broad) term “security”?
Line 362: with respect to LoRa as a communication standard: how about size of data packages that need to be transmitted? LoRa is not as good as e.g. LTE or WLAN There are – besides costs – many other aspects of communication technology that could be taken into account.
The paper then IMHO focuses far too much on communication technology and LoRaWAN. What about security? What about all the other requirements that have been extracted in the first sections?
Author Response
"Please see the attachment."

Reviewer 3 Report
The topic of the paper is very interesting and state-of-the art. Although I find it too long for a journal. It is 31 pages; more than 5 pages are the references. The list of references inherits 94 elements. This amount is already the level of a PhD thesis...
Yes, I know this paper is a literature review, although (as I wrote) I find this list way too long. Or if the focus is to show who and what did already in this field, than It should be the focus. Just literature review.
On the other hand the paper is correct, it is easy to read. I just had problems to read figures 3 and 5 where the size of the characters are too small.
Author Response
"Please see the attachment."

Round 2
Reviewer 2 Report
The comments have been taken into account and addressed; so there are only some minor comments
- from a research perspective I would be careful in using phrases such as "LoRa is ideal" (line 192) or "ideal requirements" (line 577). Please rephrase
- gendering (line 54): use "they" (= plural) instead of "him" (singular)
- 263: it is still not clear to me which databases you have been using. Please clarify
- GPS: the broader term for these positioning technologies would be GNSS. I recommend using this term
Author Response
"Please see the attachment."
